# Finite Element Analysis and Polarization Test of IDEs Piezoelectric Actuator

**DOI:** 10.3390/mi13020154

**Published:** 2022-01-20

**Authors:** Yonggang Liu, Shuliang Zhang, Aoke Zeng, Pengfei Yan

**Affiliations:** 1School of Mechatronics Engineering, Henan University of Science and Technology, Luoyang 471003, China; zak18538031886@163.com (A.Z.); yan15713883977@163.com (P.Y.); 2Collaborative Innovation Center of Machinery Equipment Advanced Manufacturing of Henan Province, Henan University of Science and Technology, Luoyang 471003, China; sl2226716700@163.com

**Keywords:** piezoelectric actuator, IDEs, FEM, polarization, electric field, strain

## Abstract

A new type of actuator is presented in the paper that integrates the IDEs into a conventional piezoelectric sheet. The electrodes and polarization play a key role in the strain. Adopting constitutive equations of piezoelectric theory and variation principles in elasticity theory, the piezoelectric component dynamic equation was deduced. Several finite element models of the IDEs piezoelectric actuator were established in ANSYS. The effect of branch electrodes on the strain of the actuator was analyzed. The results show that the strain can be bigger than that of the conventional piezoelectric sheet by decreasing the gap and increasing the width of electrodes. According to the FEM result, some IDEs piezoelectric actuators were prepared. The distribution of the static electric field inside the actuator was researched to determine the polarization voltage. The 2671 high voltage power and DU-20 temperature-controlled oil bath was applied to explore the polarization process. The effect of the voltage, time and temperature on the strain of the actuator was researched by a TF2000 and SIOS laser interferometer. The results show that the optimum polarization is 800 V, for 60 min and at 150 °C. The strain of the IDEs piezoelectric actuator is 1.87 times that of the conventional piezoelectric actuator. The actuators could prove to be helpful applications for micro-nano devices.

## 1. Introduction

The movement and positioning control of a system is accomplished by actuators. Piezoelectric actuators have the advantages of a high resolution, high frequency response, low power consumption and strong anti-electromagnetic interference ability. Therefore, they have been widely used in precision drive fields, such as microelectronic mechanical systems, microrobots, aerospace and biomedical engineering [1,2,3,4,5,6]. A precision micro-positioner using piezoelectric components as the actuators can overcome the shortcomings of conventional positioning devices and have unique advantages in micron and sub-micron precision positioning control [6,7,8,9,10]. According to previous research results, piezoelectric actuators could include conventional piezoelectric components [11,12], piezoceramic bimorphs [13,14,15], piezoelectric stacks [16,17] and composite structures, etc.

Piezoelectric ceramics has relatively large piezoelectric constants, and are often made into thin components with electrode layers covering the upper and lower surfaces, which are used directly as the actuator. Deng et al. [11] presented a planar piezoelectric actuator composed of four conventional piezoelectric elements. Through the independent movement of piezoelectric elements, the planar actuator can move gradually through the static friction forces in the non-resonant mode.

Piezoelectric bimorphs are improved piezoelectric actuators. They are made by bonding piezoelectric components and being polarized in the thickness. The metal silver is plated as an electrode on a large plane perpendicular to the thickness direction. Takata et al. [13] proposed a pseudo-piezoelectric evaluation method for the conductor, which utilized positive piezoelectric and inverse piezoelectric analyses. It shows very accurate potential distributions in the actuator and sensor modes of the thin piezoelectric bimorph with the conductor layers.

The composition principle of the piezoelectric stack is to superimpose a single piezoelectric component, and then to encapsulate it. Using the characteristic of a single piece of ceramic to stretch and deform under the action of an electric field, the total displacement reaches a considerable value, and the displacement is compared with the stack. The number of layers is proportional. Tajdari et al. [16] used a piezoelectric stack as the excitation part of the sound source, and designed a flexure mechanism as a displacement conversion mechanism.

As a driving and positioning component, the displacement of the piezoelectric actuator is relatively small, which limits its application. In order to increase the displacement of the piezoelectric component, some displacement amplification methods have been proposed. The piezoelectric component, piezoelectric bimorph and piezoelectric stack are used as the driving core. The spring component and hinge are used as the displacement amplification mechanism. The piezoelectric composite structure actuator consists of these two parts. Typical composite structures are THUNDER [18], RAINBOW [19], honeycomb [20], moonie-type [21], cymbal-type [22] and hinge-type [23,24], etc. In order to improve the displacement of the piezoelectric sheet, a macro displacement piezoelectric actuator with interdigitated electrodes (IDEs) [25,26,27,28,29,30] is proposed in this paper. The electrode structure and polarization, which affect the displacement of the actuator, are studied. The finite element method is adopted to analyze the influence of the electrode structure on the displacement of the actuator. The influence of the polarization process on piezoelectric actuator displacement is studied by a polarization test.

## 2. Actuator Structure

In essence, the driving core of the piezoelectric bimorph, piezoelectric stack and some piezoelectric composite structure actuators is still the conventional piezoelectric sheet in Figure 1. The up-and-down surface of the sheets is covered with metal electrodes. There are distinctions between the positive and negative electric charges. The polarization direction of sheet is along the Z-axis. The transverse piezoelectric effect and the *d*_31_ constant are utilized in the plane. The force and displacement in the length direction of sheets on the X-axis could be utilized. The mechanical performance of the sheets is isotropic. Thus, the strong force and displacement cannot be generated in a particular direction. Due to the fact that the piezoelectric coefficient *d*_31_ is approximately half of the *d*_33_, the strain of the sheets is relatively small.

The IDEs piezoelectric actuator can overcome these shortcomings of the conventional piezoelectric sheet. The scheme of the IDEs piezoelectric sheet is shown in Figure 2. The IDEs piezoelectric sheet is denominated by an electrical connections configuration that is made through a separate interdigitated electrode having an alternating finger-to-finger polarity. The polarization axis of the IDEs piezoelectric sheet is in the length direction. The sheet can take advantage of the relatively large piezoelectric coefficient *d*_33_ and length expansion effect, so the output displacement of the IDEs piezoelectric sheet along the length direction is larger than that of the conventional piezoelectric sheet. Under the same displacement output, the excitation voltage required by the IDEs piezoelectric sheet is relatively small, and the requirement of the sheet on the driving power supply is lower. If the IDEs piezoelectric sheet is used as the driving core, greater displacement can be generated. They could replace the conventional piezoelectric actuators in many occasions. The present study focuses on the performance of mechanics affected by electrodes and the polarization technical conditions.

## 3. Finite Element Analysis

### 3.1. Piezoelectric Component Dynamic Equation

The finite element method is a widely used and effective numerical analysis method. In particular, for structures with complex stress and strain conditions that are difficult to solve with analytical methods, the finite element method is a better choice [31,32]. Using the characteristics of standardization and normalization, the finite element method turns the large-scale analysis and calculation of complex engineering problems into reality [33,34].

The piezoelectric component is divided into elements. The *L* (Lagrange–Euler function) of the element, which comprises kinetic energy *E*_k_ and potential energy *E*_p_, can be expressed as:(1)$$L=\int_{V}\left(E_{k}-E_{p}\right)dV$$

Substituting the piezoelectric constitutive equation into Equation (1), the *L* and electric field strength can be reduced to:(2)$$\begin{array}{l} L=\int_{V}\frac{1}{2}\rho \{\dot{u_{}^{’}{\}}^{T}\{u_{}^{’}\}}dV\\ -\frac{1}{2}\int_{V}({\left\{S\right\}}^{T}[c^{E}]\{S\}-{\left\{S\right\}}^{T}[e]\{E\}-{\left\{E\right\}}^{T}[e]\{S\}-{\left\{E\right\}}^{T}[\varepsilon^{\varepsilon }]\{E\})dV \end{array}$$
where *ρ* is the mass density, {*u*’} is the speed, *V* is the volume, {*S*} is the strain, {*E*} is the electric field strength, [*c^E^*] is the stiffness matrix, [*e*] is the piezoelectric stress matrix and [*ε^ε^*] is the clamping dielectric matrix.

The virtual work of the element, including the work of the external force and electric charge, can be expressed as:(3)$$\begin{array}{l} \delta W=\int_{V}{\left\{\delta u\right\}}^{T}\{F_{b}\}dV+\int_{S_{{}^{{}_{1}}}}{\left\{\delta u\right\}}^{T}\{F_{S}\}dS_{1}+{\left\{\delta u\right\}}^{T}\{F_{p}\}\\ \;\;\;\;-\int_{V}\delta \varphi q_{b}dV-\int_{S_{2}}\delta \varphi q_{s}dS_{2}-\delta \varphi q_{p} \end{array}$$
where *δu* is the virtual displacement; *δφ* is the virtual electric potential; {*F_b_*}, {*F_s_*} and {*F_p_*} are the volume load, surface load and concentrated force, respectively; {*q_b_*}, {*q_s_*} and {*q_p_*} are the volume charge, surface charge and point charge, respectively; and *S*_1_ and *S*_2_ are the corresponding scopes.

Hamilton’s theorem establishes the relationships of *L* and the external force as follows:(4)$$\int_{t1}^{t2}\delta (L+W)dt=0$$

According to Hamilton’s theorem (4), using Equations (2) and (3), the variational equation of the element can be obtained as:(5)$$\begin{array}{l} \int_{V}(\rho \{\dot{\delta u^{’}{\}}^{T}\{u^{’}\}}-{\left\{\delta S\right\}}^{T}[c^{E}]\{S\}+{\left\{\delta S\right\}}^{T}[e]\{E\})dV\\ +\int_{V}({\left\{\delta E\right\}}^{T}[e]\{S\}+{\left\{\delta E\right\}}^{T}[\varepsilon^{\varepsilon }]\{E\}+{\left\{\delta u\right\}}^{T}\{F_{b}\}-\delta \varphi q_{b})dV\\ +\int_{S_{{}^{{}_{1}}}}{\left\{\delta u\right\}}^{T}\{F_{S}\}dS_{1}-\int_{S_{2}}\delta \varphi q_{s}dS_{2}+{\left\{\delta u\right\}}^{T}\{F_{p}\}-\delta \varphi q_{p}=0 \end{array}$$

In the finite element model, the displacement of any point can be determined by the interpolation function using the displacement of each node. Then, the balance equation and geometric equation can be expressed by the interpolation function. The interpolation equation of the load and charge are substituted into the variational equation of the piezoelectric element, and dynamic compensation is added. [*c^E^*], [*e*] and [*ε^ε^*] are the coefficients of piezoceramics. {*E*}, {*q_b_*}, {*q_s_*} and {*q_p_*} are the electrical boundary. {*F_b_*}, {*F_s_*} and {*F_p_*} are the force boundary. Assuming that these coefficients of the piezoelectric component are constants, the electrical and force boundaries are finite. The equations of the element are simplified by *δu* and *δφ*, respectively. Following element equations assembly procedures, the dynamic FEM equation of the piezoelectric component is expressed as:(6)$$\left[\begin{array}{cc} \left[M\right] & 0\\ 0 & 0 \end{array}\right]\left[\begin{array}{c} \left\{u’’\right\}\\ 0 \end{array}\right]+\left[\begin{array}{cc} \left[C^{f}\right] & 0\\ 0 & 0 \end{array}\right]\left[\begin{array}{c} \left\{u’\right\}\\ 0 \end{array}\right]+\left[\begin{array}{cc} \left[K_{uu}\right] & \left[K_{u\varphi }\right]\\ \left[K_{\varphi u}\right] & \left[K_{\varphi \varphi }\right] \end{array}\right]\left[\begin{array}{c} \left\{u\right\}\\ \left\{\varphi \right\} \end{array}\right]=\left[\begin{array}{c} \left\{F\right\}\\ \left\{Q\right\} \end{array}\right]$$
where [*M*] is the mass matrix, [*C*_f_] is the damp matrix, [*K*_uu_] is the elastic matrix, [*K*_uφ_] and [*K*_φu_] are the piezoelectricity matrix, [*K*_φφ_] is the dielectric coefficient matrix, {*F*} is the force and {*Q*} is the charge. It is noted that there is a part about the charge in the piezoelectric component equation while [*K*_uφ_] and [*K*_φu_] reflect the coupling relationship between the mechanical properties and the dielectric properties of piezoelectric body.

[*M*], [*C*_f_] and [*K*_uu_] are constant matrices, {*F*} is a finite distributed force vector and {*Q*} is related to the electric field strength. Therefore, the electrode of the component and the applied voltage play a decisive role in {*Q*}. [*K*_uφ_] and [*K*_φu_] are related to the remanent polarization of the component. [*K*_φφ_] is a complex coefficient, which is determined by the remanent polarization, piezoelectric material and electrode structure. The polarization process and the electric field strength when the component is polarized determines the remanent polarization of the component, so they determine [*K*_uφ_], [*K*_φu_] and [*K*_φφ_]. The polarization process is a key factor affecting the actuator performance.

The electric field strength of the conventional surface electrode piezoelectric component is uniform. The electric field intensity distribution of the IDEs piezoelectric component is nonlinear. The structure of IDEs affects the distribution of the electric field strength. Therefore, it affects the electrical and dielectric coefficients of Equation (6). IDEs are another key factor that affects the component performance. Due to the uneven electric field distribution, {*Q*} inside the piezoelectric component is a function of position (*x*,*y*,*z*). Therefore, it is more difficult to establish and solve the system dynamic Equation (6) through Equation (5).

### 3.2. Mechanical Analysis

#### 3.2.1. Finite Element Model

Finite element software ANSYS was adopted to analyze the strain and stress of the piezoelectric component. The piezoelectric material is PZT-5. The IDEs piezoelectric component is 56 × 15 × 0.5 mm^3^ in size. In the following analysis, the IDEs piezoelectric component is assumed to be fully polarized. The density is 7.45 × 10^3^ kg/m^3^. The elastic compliance constant *S^E^*_11_ is 15 × 10^−12^ m^2^/N, *S^E^*_33_ is 19 × 10^−12^ m^2^/N and *S^E^*_55_ is 22 × 10^−12^ m^2^/N. The piezoelectric constant *d*_31_ is −185 × 10^−12^ C/N, *d*_33_ is 400 × 10^−12^ C/N and *d*_15_ is 650 × 10^−12^ C/N. The dielectric constant *ε^T^*_31_ is 2400 and *ε^T^*_33_ is 2100. On both sides of the center line of the branch electrode, the polarization direction of the piezoceramic is opposite. During the finite element analysis, two piezoelectric material models need to be established. Model 1: The polarization direction is parallel to the X-axis and points to the positive direction of the X-axis. Model 2: The polarization direction is parallel to the X-axis and points to the negative direction of the X-axis. Before meshing the finite element model, adjacent piezoceramics were set to different piezoelectric material models. The IDEs piezoelectric component was modeled with 320,000~550,000 SOLID98 ELEMENTS. The finite element model of the component is shown in Figure 3.

#### 3.2.2. Analysis Results and Discussion

Actuators vibrate when the varying voltages are applied to the IDEs piezoelectric component. The free deformation of the component when a static voltage of 200 V was applied to the model is shown in Figure 4. As can be seen, the left end is the X-axis zero point, and the right end face of the component produces obvious X-direction elongation displacement. At the same time, the lower end is positioned as the Y-axis zero point, and the upper end face of the component produces Y-axis compression displacement. The analysis shows that the components have orthorhombic anisotropy.

The width of the branches is 0.5 mm. The X-axis displacements of the nodes on surface III are extracted and averaged. The lengthways strain of the sample can be obtained from the average displacement. The Y-axis displacements of the nodes on surface IV are extracted and averaged. The transverse strain can be obtained from the displacement. The strains versus the gap of electrodes are shown in Figure 5. It can be seen that the non-linear displacement of the component decreases with the increase in the electrode gap. When the gap is less than 1 mm, the curve drops faster. When the gap is more than 1 mm, the curve tends to be flat. The main reason is that, as the gap increases, the average value of the electric field strength becomes smaller. The induced strain of the component becomes smaller accordingly. Therefore, when preparing the IDEs piezoelectric component, the gap should be made smaller if the electrode manufacturing process permits it.

The gap between each pair of branches is 0.5 mm. The lengthways and transverse strains versus the width are shown in Figure 6. The strain increases as the electrode width increases. It can be seen from Figure 5 and Figure 6 that the electrode width has a relatively small effect on the strain of the component.

The longitudinal elongation strain is greater than the transverse contraction strain, and the increase in the rate of the longitudinal strain is also greater than that of the lateral strain. This is because the longitudinal strain uses the piezoelectric constant *d*_33_, and the transverse strain uses the piezoelectric constant *d*_31_. The *d*_33_ of the piezoelectric material is greater than *d*_31_. The directions of the longitudinal strain and transverse strain in the plane are opposite, indicating that the component has orthorhombicity. This is caused by the Poisson effect.

Analyzing the conventional piezoelectric actuator of the same sizes and material, the strain obtained is 37 micro-strains. According to Figure 5 and Figure 6, it can be seen that the max strain of the IDEs piezoelectric actuator is 53 micro-strains. Therefore, it is easy to choose the right electrode with a smaller gap and a larger width, and the displacement of the IDEs piezoelectric actuator can exceed that of the conventional piezoelectric actuator.

### 3.3. Electric Field Analysis

The polarization electric field of PZT-5 is approximately 3~5 kV/mm. Before the sample is polarized, the polarization voltage needs to be determined. The polarization of the piezoelectric sample is related to the applied DC electric field. However, the electric field of the IDEs piezoelectric sample is uneven. The electric fields should be analyzed to choose the voltage. In the direction of the length of the piezoelectric actuator, a two-dimensional surface between a pair of branch electrodes was extracted as a research object. The width is 0.5 mm and the gap is 0.25 mm. A representative lengthways section model between adjacent electrodes is built in ANSYS, and the electric field is analyzed with PLANE121 elements. The number of nodes is 4497 and the number of units is 1716. The electric field is shown in Figure 7. It is not uniform. The electric field strength near the electrode is relatively large. The electric fields have a relatively large arc. In the middle of the section model, the electric field strength is the smallest. The electric fields are more horizontal. With an increase in the gap, the thin area of electric fields decreases. This phenomenon can also explain the decrease in the absolute value of the strain in Figure 5. With an increase in width, the dense area of electric field lines increases. This phenomenon can also explain the increase in the absolute value of the strain in Figure 6.

The maximum electric field strength appears at the mark points MAX in the section model. The direction cosine of the maximum electric field is (0.612, −0.790). It does not change with changes to the external voltage. Figure 8 shows the maximum electric field strengths of the IDEs piezoelectric actuator, which are on the left-hand side of the vertical axes. The electric fields are uniform if the voltages are applied in the 1 mm-thick conventional piezoelectric actuator. They are shown on the right-hand side of the vertical axes. While the voltage increases, the internal electric field strength of the medium increases linearly. This result is consistent with the Gaussian electrostatic field theorem. It can be seen that the maximum electric field strength exceeds the dielectric breakdown strength (5.0~7 kV/mm) when the external voltage is 900 V. However, the electric field strength of the conventional piezoelectric sample, which has the same size as the IDEs piezoelectric sample, is only 0.9 kV/mm at 900 V. The conventional piezoelectric sample cannot be polarized at this voltage. Therefore, the IDEs piezoelectric sample can be polarized at the low voltage, and the low-cost power supply is needed. In addition, the safety is relatively large in polarization.

## 4. Polarization Experiment

### 4.1. Preparation and Polarization Process

According to the results of the finite element analysis, some samples were prepared, with a length of 56 mm, a width of 15 mm and a thickness of 1 mm. The width of the electrode is 0.5 mm and the gap of the electrodes is 0.25 mm. The IDEs piezoelectric sample is shown in Figure 9.

After the sample is prepared, it has no piezoelectric effect. Piezoelectric polarization is essentially an ordered arrangement of the electric domain orientation in the polarization direction. Before polarization, the crystal grains inside piezoelectric ceramics are disordered and irregular due to their own polarization directions. The sample has no external piezoelectric performance. The distribution of the electric domain before polarization is shown in Figure 10a. The piezoelectric sample is applied in a certain high DC field and temperature. The spontaneously polarized electrical domains are arranged in an orderly and regular manner. They are consistent with the direction of the electric field. The distribution of the polarization process is shown in Figure 10b. After polarization, most of the electrical domains follow the direction of the DC electric field, so the piezoelectric ceramic retains a certain polarization strength. The distribution of the electric domain after polarization is shown in Figure 10c. The piezoelectric sample can show a polarity, anisotropy and piezoelectric effect after the polarization. It has a great impact on the piezoelectric properties of the piezoelectric sample, regardless of whether they were adequate polarized. The polarization conditions include the temperature, electric field and time. They play important roles in the polarization of the piezoelectric sample. To make the sample completely polarized and draw out their potential piezoelectric properties, the polarization conditions must be reasonable.

### 4.2. Experiments

The sample was polarized by thermal polarization. The main feature of the thermal polarization is that the polarization temperature is lower than the Curie temperature. The polarization device is simple to operate. The polarization effect is greater too. The 2671 High Voltage Power Supply and a DU-20 Temperature-Controlled Oil Bath was adopted in polarization. The polarized equipment is shown in Figure 11. The 2671 High Voltage Power Supply provides DC high voltage. The DU-20 Temperature-Controlled Oil Bath fills with methyl silicone oil, which heats the silicone oil and keeps it warm. The sample was placed in methyl silicone oil and subjected to oil bath polarization. This avoids the mental electrode being oxidized. The sample was then uniformly heated.

The polarization process is as follows: Check the sample with an ammeter and exclude samples with a resistance value less than 108 Ω; wash the sample with acetone and dry it; mount the sample on the polarization fixture to ensure that the electrode of the sample is in effective contact with the positive and negative electrodes of the power supply; turn on the oil bath and heat the silicone oil to the polarization temperature; place the polarization fixture and sample into the oil bath for 10 min to bring the sample to the polarization temperature; slowly apply voltage to the polarization voltage; maintain a certain time at the polarization voltage and temperature; turn off the high voltage power supply; quickly remove the polarization fixture and sample and place them at room temperature with methyl silicone oil; quickly apply a voltage that is slightly lower than the polarization voltage; after the sample is cooled to room temperature, turn off the high voltage power supply; wash the sample with acetone and wipe the acetone off with a cotton wool; allow for the sample to dry naturally at room temperature for 24 h.

### 4.3. Results and Analysis

#### 4.3.1. The Influence of Voltage

Micro-displacement is measured by a TF2000 and SIOS Laser Interferometer. It is shown in Figure 12. The laser directly strikes the length end of the sample in Figure 13. Different polarization voltages are imposed for 20 min at 110 °C. The induced strain along the length direction versus the polarization voltage is shown in Figure 14. There is obvious hysteresis in the IDEs piezoelectric sample. The curve is asymmetric about the origin center of the voltage coordinate. It can be seen that the peak strain of the sample increases with the increase in the polarization voltage in the same test conditions. The peak strain increases tardily when the DC voltage varies from 750 V to 800 V. When the voltage exceeds 800 V, the sample is often broken down during polarization.

#### 4.3.2. The Influence of Temperature

Polarization experiments were carried out on samples using different temperatures for 20 min at 500 V. The effect of the polarization temperature on the induced strain of the sample is shown in Figure 15. It can be seen that, under the same test conditions, the peak of the induced peak strain of the sample increases as the polarization temperature increases. In general, as the temperature increases, the molecular thermal motion of the sample becomes more active. The dipole moment of spontaneous polarization is easier to rotate. Under the action of an external DC high electric field, the dipole moment of spontaneous polarization is more easily oriented and it is easy to polarize the sample. However, the temperature of the polarization is too high and the insulation of the sample is lowered. The dielectric breakdown strength is lowered, which tends to cause the sample to be broken down. According to the experiment, when the polarization temperature rises to 150 °C, there is no large increase in the peak-induced strain of the sample, and the sample is easily broken down in polarization.

#### 4.3.3. The Influence of Time

The polarization time refers to the holding pressure and time required for the crystal inside the sample to change from one equilibrium state to another. The response frequency of atomic polarization is approximately 10^13^ Hz, whereas the response frequency of electron polarization is approximately 10^15^ Hz. After the polarization voltage is loaded, the polarization of electrons and atoms is completed in an instant. However, it takes time to reverse and reorient the domains. If the polarization time is not long enough and the polarization voltage is not high, the orientation of the domains could not be good enough. Due to the limitation of the dielectric breakdown strength, the polarized DC electric field of the sample cannot be too high. Therefore, it is very important to extend the polarization time.

The sample is polarized at a polarization voltage of 500 V and a polarization temperature of 110 °C. The effect of the polarization time on the induced strain of the sample is shown in Figure 16. It can be seen that, under the same conditions of the polarization voltage and polarization temperature, the peak of the induced strain of the sample increases with the increase in the polarization time. When the polarization time reached 60 min, there was no significant change in the induced strain of the sample.

In general, the longer the polarization time, the more gradually the domain overcomes various resistances. The higher the degree of the alignment of the domains along the direction of the external electric field, the better the polarization effect. This is because the initial polarization is the inverse of 180 domains, and, finally, the steering of 90 domains. The reversal of the 180 domains generally does not cause internal gravity and can be realized very quickly. The 90-domain steering is difficult to perform due to stress. When the polarization process exceeds a certain period of time, the alignment of the domains has been completed. Extending the polarization time does not have much effect on the degree of the polarization of the sample. The final piezoelectric properties of the sample will not vastly increase.

According to the experiment, the optimal polarization process conditions are: the polarization voltage is 800 V, the polarization temperature is 150 °C and the polarization time is 60 min. The piezoelectric actuator is 56 mm long × 15 mm wide × 1 mm high. The electrode measures 0.5 mm width × 0.25 mm gap. The strain is shown in Figure 17. The strains of the conventional piezoelectric actuator with the same size are shown too. It can be seen that the peak strain of the IDEs piezoelectric actuator is 84 micro-strains, which is 1.87 times that of the conventional piezoelectric actuator’s 45 micro-strains. The rising section fluctuates because the preparation accuracy of the electrodes is not high. The polarization of the conventional piezoelectric actuator often requires a higher voltage. Therefore, the IDEs piezoelectric actuator has lower requirements for polarization equipment. The IDEs piezoelectric actuator should be better in its displacement output and demand for process equipment. Comparing Figure 6 and Figure 17, it can be seen that the peak strain is larger than that (53 micro strain) of the finite element analysis. This is because the piezoelectric and dielectric coefficients used in the finite element analysis are the standard values of PZT-5, which are relatively small. The polarization process summarized above can make the IDEs piezoelectric actuator more fully polarized, so as to obtain a greater peak strain.

## 5. Conclusions

The structural characteristics of the IDEs piezoelectric actuator are analyzed. Adopting constitutive equations of piezoelectric theory and variation principles of elasticity theory, the finite element equation of the piezoelectric component was obtained. The model of the IDEs piezoelectric actuator was established by ANSYS, and the influence of the electrode gap and width on the strain was analyzed. The mechanical analysis shows that the actuator has orthotropic properties. The stain decreases with the increase in the gap. As the width becomes larger, the strain in the longitudinal direction increases. In order to determine the polarization electric field, a planar electric field analysis was performed on the actuator. The variation of the maximum electric field between adjacent electrodes of the actuator was obtained. A PZT-5 IDEs piezoelectric actuator was prepared based on the results of the finite element analysis. A polarization experimental platform was established to polarize the sample. After polarization, the sample was subjected to a displacement measurement. The experiment has obtained the influence of the voltage, temperature and time on the displacement of the actuator. The results show that, with an increase in the polarization voltage, polarization temperature and polarization time, the displacement of the actuator increases. However, an excessive polarization voltage and polarization temperature tend to cause the sample to be broken down. When the polarization time exceeds a certain level, the displacement of the actuator does not change significantly. The best polarization process conditions are: 800 V, 150 °C and 60 min. The strain of the IDEs piezoelectric actuator is 1.87 times that of the conventional piezoelectric actuator. It can replace conventional piezoelectric actuators in the field of precision drives, and has great application prospects.

## Figures and Tables

**Figure 1 micromachines-13-00154-f001:**
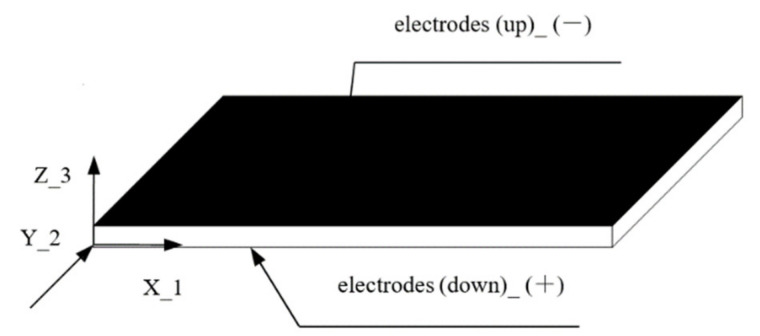
Model of conventional piezoceramic sheet.

**Figure 2 micromachines-13-00154-f002:**
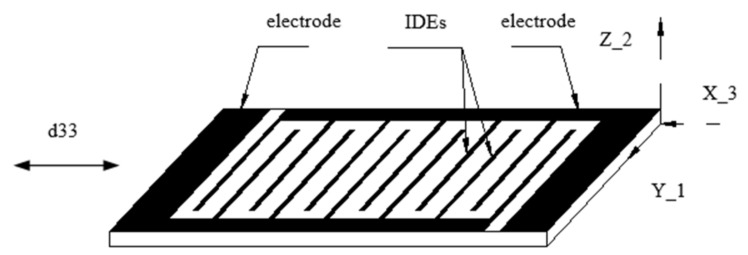
Model of IDEs piezoceramic sheet.

**Figure 3 micromachines-13-00154-f003:**
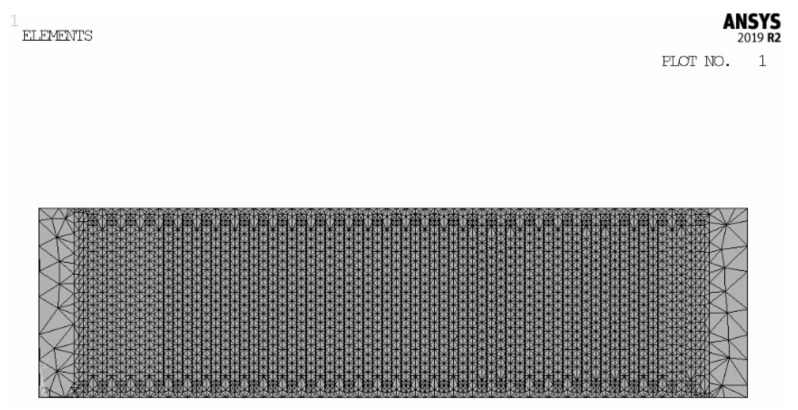
Finite element model.

**Figure 4 micromachines-13-00154-f004:**
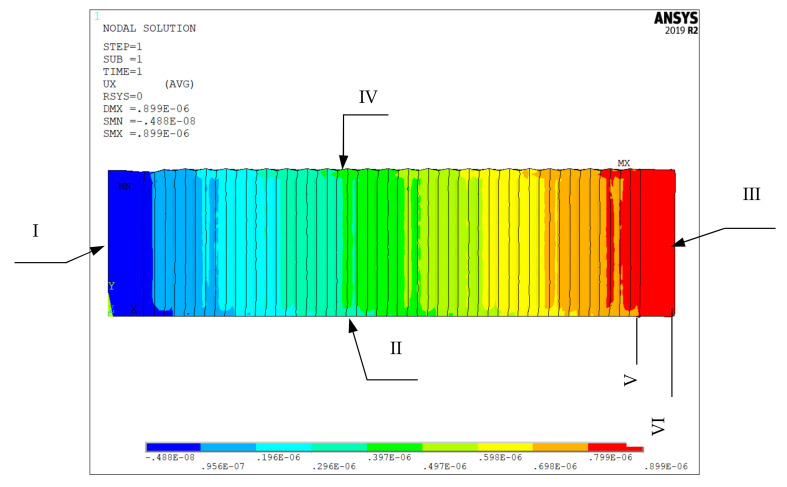
Displacement. I is the positioning surface of X axis; II is the positioning surface of Y-axis; III and IV are free surface; V is the end line before the x-direction deformation and VI is the end line after the deformation.

**Figure 5 micromachines-13-00154-f005:**
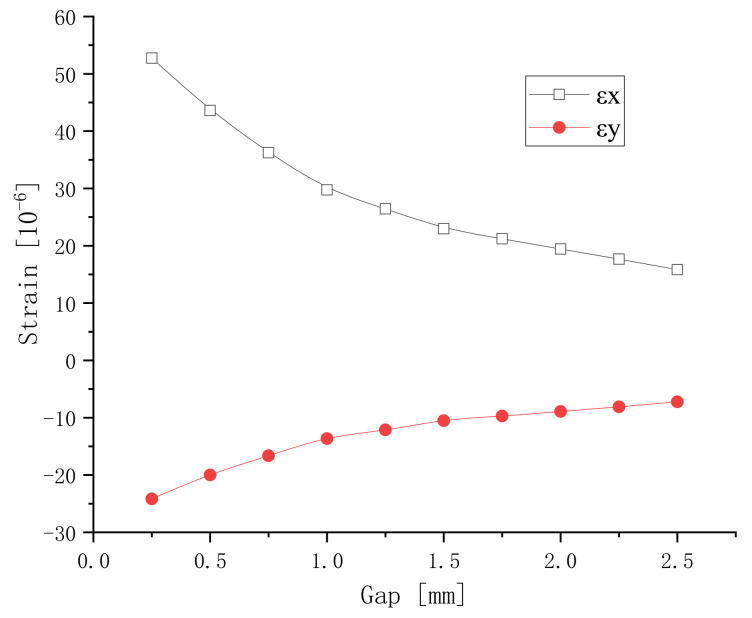
Lengthways and transverse strains versus the gap.

**Figure 6 micromachines-13-00154-f006:**
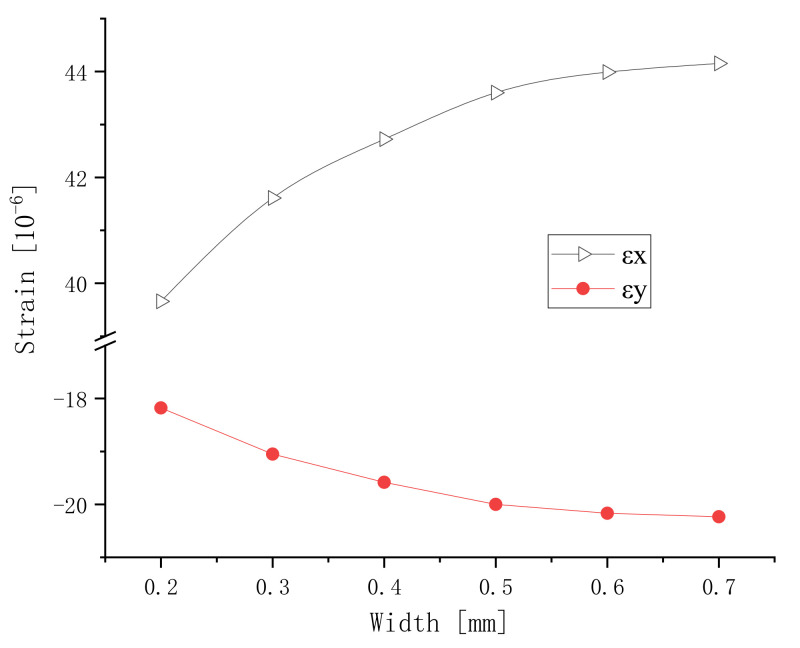
Lengthways and transverse strains versus the width.

**Figure 7 micromachines-13-00154-f007:**
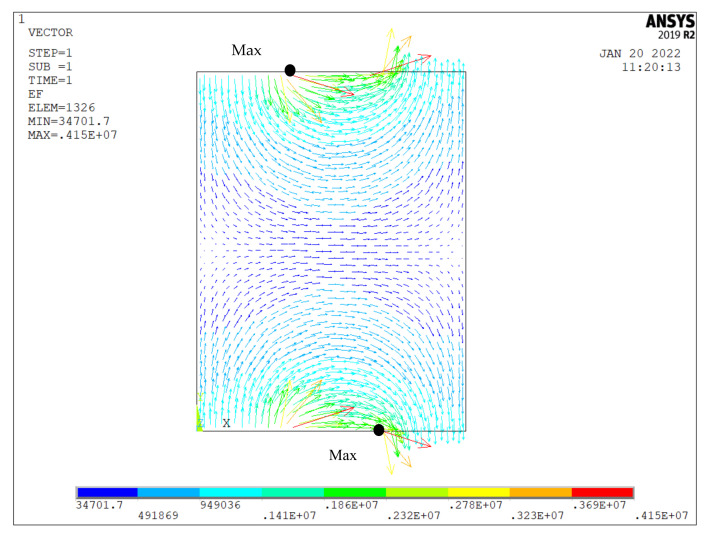
Electric field of section model.

**Figure 8 micromachines-13-00154-f008:**
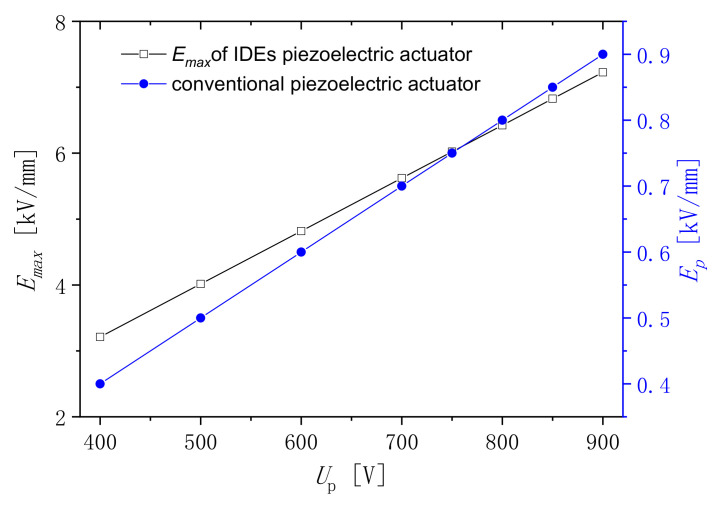
Maximum electric field.

**Figure 9 micromachines-13-00154-f009:**
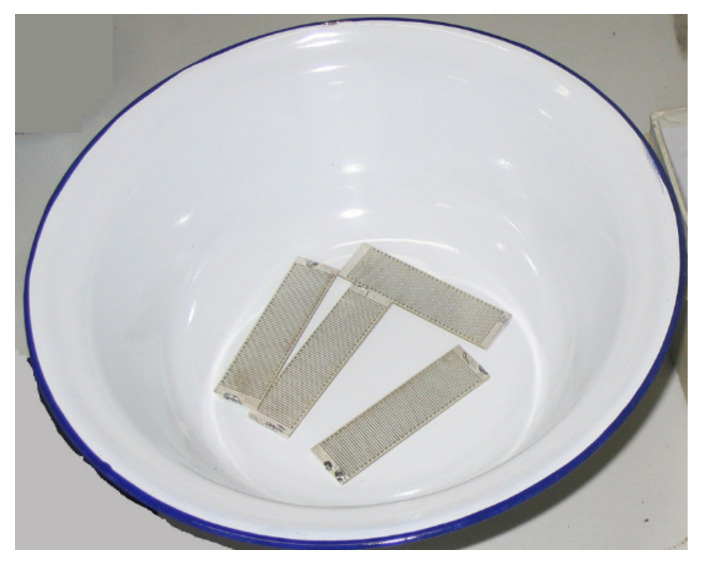
IDEs piezoelectric sample (PZT-5).

**Figure 10 micromachines-13-00154-f010:**
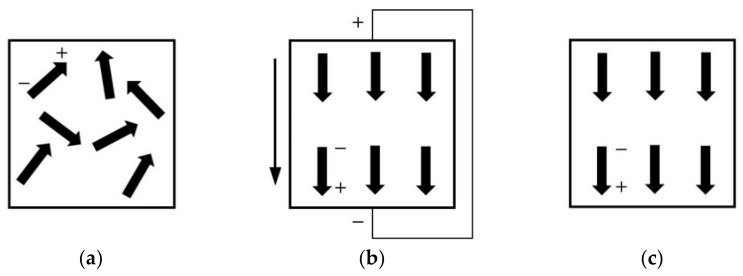
Principle of polarization: (**a**) before; (**b**) polarization; (**c**) after.

**Figure 11 micromachines-13-00154-f011:**
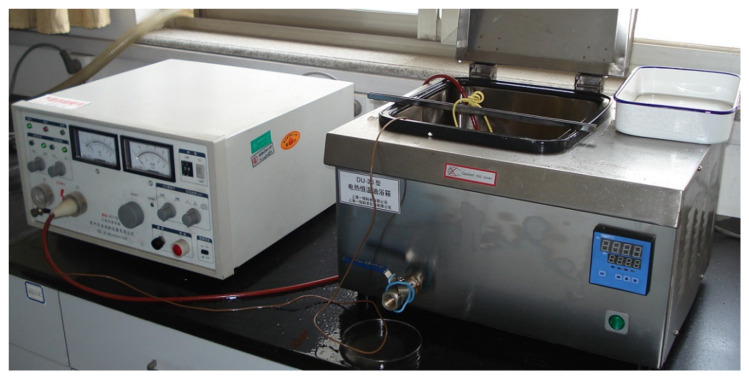
Polarization equipment.

**Figure 12 micromachines-13-00154-f012:**
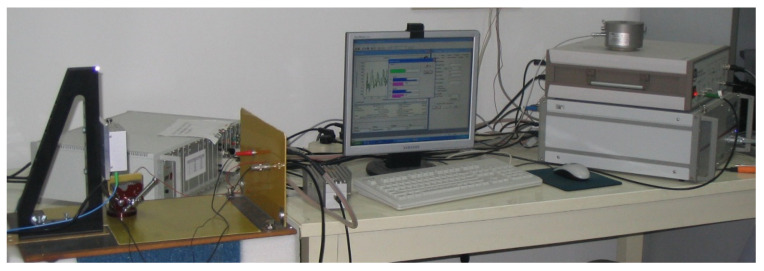
Measurement instruments.

**Figure 13 micromachines-13-00154-f013:**
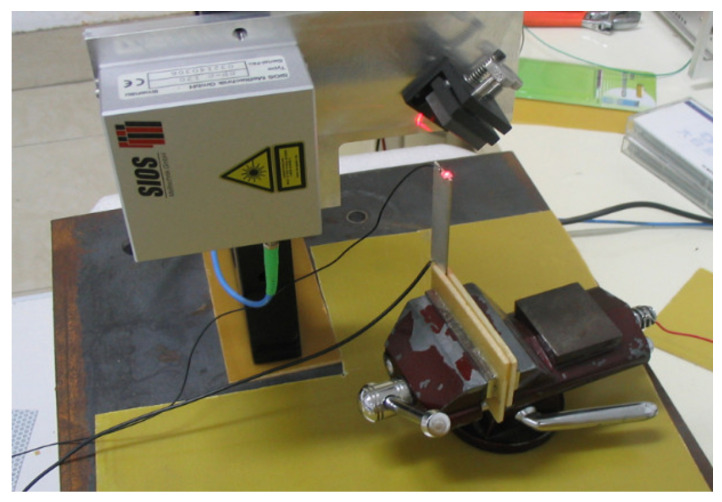
Laser head and actuator.

**Figure 14 micromachines-13-00154-f014:**
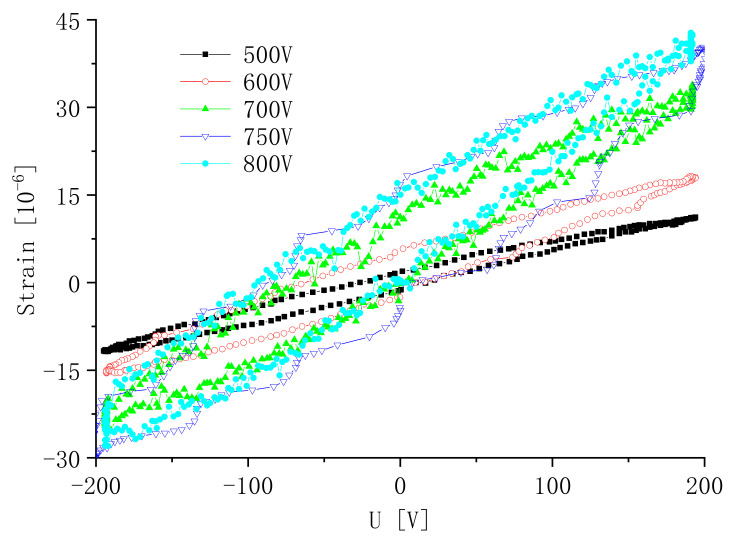
Strain versus the polarization voltage.

**Figure 15 micromachines-13-00154-f015:**
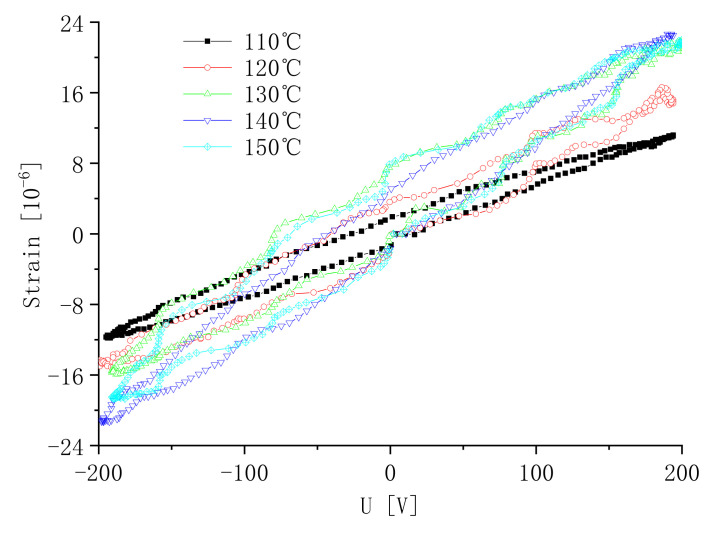
Strain versus the polarization temperature.

**Figure 16 micromachines-13-00154-f016:**
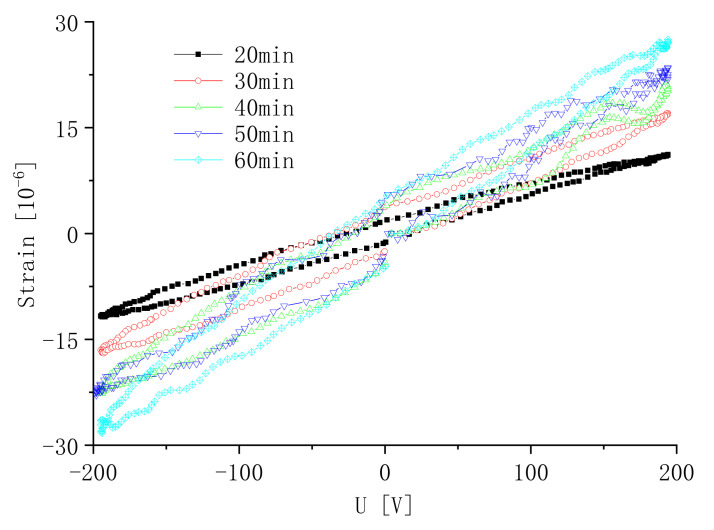
Strain versus the polarization time.

**Figure 17 micromachines-13-00154-f017:**
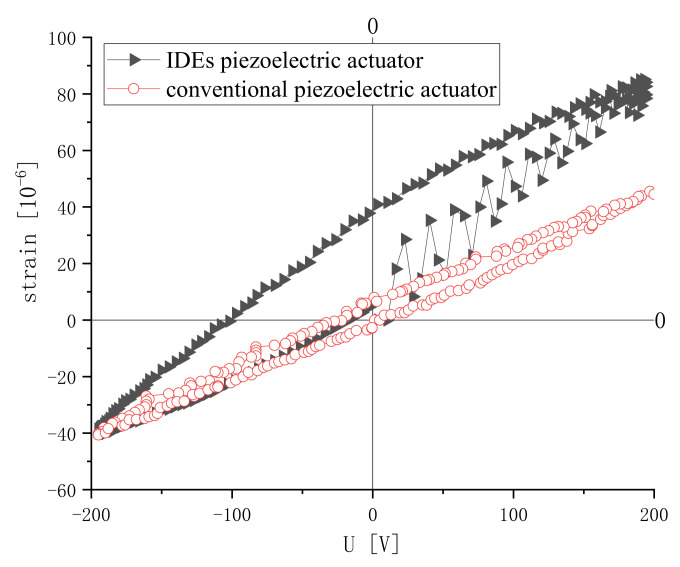
Strain of IDEs piezoelectric actuator and conventional piezoelectric actuator.

## Data Availability

Data are available upon request from the corresponding author.

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
