# Peer review of "Finite Element Analysis and Polarization Test of IDEs Piezoelectric Actuator"

_micromachines, 2022, doi:10.3390/mi13020154_

Round 1

Reviewer 1 Report

Dear Authors,

This paper needs revision before it can be published. I strongly suggest  English and grammatic check again. Please pay attention to the rules of academic paper.

My comments are listed in the attachment.  

Kind Regards

Author Response

Response to Reviewer 1 Comments

Point 1: Lines 22: “time” should be “times”. Grammatic mistake.

Response 1: It has been changed.

Point 2: Lines 27-28: Something missing in this sentence. “Piezoelectric have” should be either “Piezoelectric motors have” or “Piezoelectric stages have”. Please correct it.

Response 2: Piezoelectric actuators have the advantages of high resolution,

Point 3: Lines 29: Grammatic mistake. “it has been” should be changed with “they have been”

Response 3: It has been changed.

Point 4: Lines 31: Please add following basic Reference about applications in [1-6]:

[xx] C. Zhao, Ultrasonic Motors Technologies and Applications, Berlin, Germany: Springer, 2011.

Response 4: The reference has been added.

Point 5: Lines 34: Please add more recent publications about precision position control like the ones below:

[xx] Peng-Zhi Li, Xiao-Dong Wang, Lei Zhao, De-Fu Zhang, Kang Guo, Dynamic linear modeling, identification and precise control of a walking piezo-actuated stage, Mechanical Systems and Signal Processing, Volume 128, 2019, pp. 141-152.

[xx] Makarem, S.; Delibas, B.; Koc, B. Data-Driven Tuning of PID Controlled Piezoelectric Ultrasonic Motor. Actuators 2021, 10, 148.

Response 5: The two references have been added.

Point 6: Line 39: “et al” should be “et al.”

Response 6: It has been changed.

Point 7: Line 45: “et al” should be “et al.”

Response 8: It has been changed.

Point 9: Lines 46-47: “for the piezoelectric and inverse-piezoelectric analyses” is not clear for me. Do you mean “for the direct-piezoelectric and inverse-piezoelectric analyses”?

Response 9: Lines 46-47: Takata et al. proposed a pseudo-piezoelectric evaluation method for the conductor is for the positive-piezoelectric and inverse-piezoelectric analyses.

Point 10: Line 53: “et al” should be “et al.”

Response 10: It has been changed.

Point 11: Line 59: “stack is used “should be “stack are used”. Grammatic mistake

Response 11: It has been changed.

Point 12: Line 65: Your manuscript is mainly about IDEs piezoelectric actuator. But you mention only two references about it. Two references are not adequate for the state of art technology. I strongly recommend you to add more references about IDEs. I have some in followings. Please add them;

[xx] Han, L.: Zhao, H.: Xia, H.: Pan, C.: Jiang, Y., Li, W.: Yu, L.: A Compact Impact Rotary Motor Based on a Piezoelectric Tube Actuator with Helical Interdigitated Electrodes, Sensors, MDPI, vol. 18, 2018

[xx] Fuda, Y.: Yoshida, T.: Piezoelectric torsional actuator, Ferroelectrics, vol. 160, 1994, pp. 323-330

[xx] M. Wischnewski, B. Delibas and A. Wischnewski, "Piezoelectric motor operated both in resonance and DC modes," ACTUATOR; International Conference and Exhibition on New Actuator Systems and Applications 2021, 2021, pp. 1-4.

[xx] Bowen, C. R.: Nelson, L. J.: Stevens, R.: Cain, M. G.: Steward, M.: Optimization of interdigitated electrodes for piezoelectric actuators and active fiber composites, J. Electro Ceram, vol. 16, 2006, pp. 263- 269

Response 12: Those references have been added.

Point 13: Lines 76: “The sheets output force and displacement in the length direction “. This sentence grammatically wrong. Please correct it.

Response 13: The force and displacement in the length direction of sheets on X-axis could be utilized.

Point 14: Line 110: {u`} is not same with the one in equation 2. Please use same variables. Be careful about the commas over it.

Response 14: It has been changed.

Point 15: Line 144: “polarized determine” has to be “polarized to determine”

Response 15: It has been changed.

Point 16: Line 150: “IDEs is” is IDEs plural or singular? When it is plural, please use “IDEs are another key factor that affect component performance”

Response 16: It has been changed.

Point 17: Line 158: “kg /m2” Unit of density must be “kg /m3”. Please correct it

Response 17: It has been changed.

Point 18: Figure 5: “Gap (mm)” in X-axis should be “Gap [mm]”.

Response 18: It has been changed.

Point 19: Figure 6: “Width (mm)” in X-axis should be “Width [mm]”.

Response 19: It has been changed.

Point 20: Line 222:” The piezoelectric”. Do you mean “The piezoelectricity”? Please modify the sentence.

Response 20: Line272: The piezoelectric sample is applied in certain high DC field and temperature.

Point 21: Line 229: “It is a great impact on the properties of the piezoelectric sample whether adequate polarization.” This sentence is not understandable. Please reformulate it.

Response 21: Line279: It is a great impact on the piezoelectric properties of the piezoelectric sample whether they were adequate polarized.

Point 3: Line 241: “In order to polarize the piezoelectric, the piezoelectric is first subjected to electric field analysis". Something missing with this sentence. Please modify it.

Response 4: Line 231: The electric fields should be analyzed to choose the voltage.

Point 3: Line 249: “The electric fields more horizontal “should be “The electric fields are more horizontal”

Response 4: It has been changed.

Point 3: Figure 10: Unit of electric field in the vertical axes has to be changed. You wrote “[Kv/mm]” Please correct them. They should be “[kV/mm]”.

Response 4: It has been changed.

Point 3: Figure: 10: How did you find the values of IDEs piezo element on the right-hand side of vertical axes? Please explain it.

Response 4: Line 249-251: Figure 8 shows the maximum electric field strengths of IDEs piezoelectric actuator that are on the left-hand side of vertical axes. The electric fields are uniformity if the voltages are applied in 1 mm thick conventional piezoelectric actuator. They are shown on the right-hand side of vertical axes.

Point 3: Line 264: “is only 0.09kV/mm in 900V” Are you sure about that? My calculation is 0.9kV/mm if you apply 900 V in 1 mm thick piezo element. Please check it again. Figure 10 has to be changed accordingly.

Response 4: It has been changed to 0.9kV/mm.

Point 3: Line 300: “20min in1100°C “should be ”20 min in 1100 °C”

Response 4: It has been changed.

Point 3: Line 300:” induce” should be “induced”

Response 4: It has been changed.

Point 3: Line 314: “20min at 500V” should be “20 min at 500 V”

Response 4: It has been changed.

Point 3: Line 342: “Figure 16”. Why is it red colored? Please change to black.

Response 4: It has been changed.

Point 3: Reference 6: “piezoelectric2“should be “piezoelectric 2“. Put space between “piezoelectric” and “2”.

Response 4: It has been changed.

Reviewer 2 Report

In this research paper, the authors designed and fabricated a new actuator that is capable to be integrated with IDEs to improve the displacement of the convention piezoelectric sheet. The authors provided both the numerical simulation and experimental results. The manuscript seems to be mildly novel and of interest to Journal of Micromachines readers, however, the overall quality of the manuscript needs minor improvement before being accepted for publication. Some of these minor suggestions are listed as follows:

  • Figure 1 shows a model for the conventional piezoceramic sheet, but it does not provide any details about the positive and negative surfaces. Also, where are the thickness and length directions?
  • The authors stated that “The stress and strain of the structure can be obtained more accurately.” Could you please provide a percentage showing the maximum and allowable accuracy using the developed FEM?
  • The authors have not accounted for the damping effect in the EOM. Any clarifications?
  • Figure 3 shows the meshing element used in the FEM; however, the figure is not clear, and no information can be extracted out of it.
  • The authors do not mention any information about the sheet boundary conditions and motion constraints (section 3.3).
  • More details are required to explain the displacement presented in Figure 4.
  • The authors stated that “When the gap is more than 1mm, the curve tends to be flat. The main reason is that as the gap increases, the average value of the electric field strength becomes smaller.” I believe the electric field is saturated when the electrode gap increases.
  • Figure 5 shows the strain versus the gap. Where are you are taking these readings on the sheet? It is not clear.
  • Figure 6 has a similar issue to that presented in Figure 5.
  • It is highly recommended to show the piezoelectric constants d33 and d31 on a figure so the readers can pick the strain direction easily.
  • As a suggestion for the authors, it will be more organized if you move Section 4.2 to the Finite Element analysis. It does add anything for the sample’s preparation and experimental procedure.  
  • “The easier the sample is polarized”. Re-write this statement. It can’t stand alone.
  • Change the color of Figure 16 in line 342.
  • As the polarization time reached 60 minutes, there was no significant change in the induced strain of the sample. Could provide another reason in addition to the resistance?
  • Change Discussion to Conclusion and revise the text to reflect the work done in the manuscript.
  • There are a few typos in the manuscript. Please revise it carefully. 

Author Response

Point 1: Figure 1 shows a model for the conventional piezoceramic sheet, but it does not provide any details about the positive and negative surfaces. Also, where are the thickness and length directions?

Response: Line 76: “The polarization direction of sheet is along the Z-axis.”

Line 78: “The force and displacement in the length direction of sheets on X-axis could be utilized.”

Point 2: The authors stated that “The stress and strain of the structure can be obtained more accurately.” Could you please provide a percentage showing the maximum and allowable accuracy using the developed FEM?

Response: Line102: “Using the characteristics of standardization and normalization, the finite element method turns the large-scale analysis and calculation of complex engineering problems into reality.”

Two references on FEM are added in Line105.

Point 3: The authors have not accounted for the damping effect in the EOM. Any clarifications?

Response: The actuator is a single piezoelectric component without other damping materials. In this paper, the free strain of the driver is studied, and the influence of the base is not considered, so there is no damping effect.

Point 4: Figure 3 shows the meshing element used in the FEM; however, the figure is not clear, and no information can be extracted out of it.

Response: Figure 3 is Responsed as:

Point 5: The authors do not mention any information about the sheet boundary conditions and motion constraints (section 3.3).

Point 6: More details are required to explain the displacement presented in Figure 4.

Response: Figure 4 is revised as:

Line184: The I is the positioning surface of X axis; the II is the positioning surface of Y axis; the III and IV are free surface; the V is the end line before the x-direction deformation and the VI is the end line after the deformation.

Point 7: The authors stated that “When the gap is more than 1mm, the curve tends to be flat. The main reason is that as the gap increases, the average value of the electric field strength becomes smaller.” I believe the electric field is saturated when the electrode gap increases.

Response: The electric field intensity includes a sparse region in the middle and a nonlinear dense region near the electrode at both ends in figure 7. The maximum electric field is in the region near the electrodes. Therefore, the electric field of IDEs samples is not easy to be saturated.

Point 8: Figure 5 shows the strain versus the gap. Where are you are taking these readings on the sheet? It is not clear. Figure 6 has a similar issue to that presented in Figure 5.

Response: Line186: The X-axis displacements of nodes on surface III are extracted and averaged. The lengthways strain of the sample can be obtained from the average displacement. The Y axis displacements of nodes on surface IV are extracted and averaged. The transverse strain can be obtained from the displacement. The strains versus gap of electrodes are shown in Figure 5.

Point 9: It is highly recommended to show the piezoelectric constants d33 and d31 on a figure so the readers can pick the strain direction easily.

Response: Line81 and Line96:Polarization axises 1-2-3 are marked in Figures 1 and 2.

Point10: As a suggestion for the authors, it will be more organized if you move Section 4.2 to the Finite Element analysis. It does add anything for the sample’s preparation and experimental procedure.  

Response: Line217:Section 4.2 was adjusted to Section 3.3

Point11: “The easier the sample is polarized”. Re-write this statement. It can’t stand alone.

Response: Line 378:Under the action of an external DC high electric field, the dipole moment of spontaneous polarization is more easily oriented and the sample is easy to be polarized.

Point12: Change the color of Figure 16 in line 342.

Response: It has been changed.

Point13: As the polarization time reached 60 minutes, there was no significant change in the induced strain of the sample. Could provide another reason in addition to the resistance?

Response: Temperature stability and impedance of IDEs actuators are being studied.

Point14: Change Discussion to Conclusion and revise the text to reflect the work done in the manuscript.

Response: It has been changed.

Point15: There are a few typos in the manuscript. Please revise it carefully. 

Response: Some typos has been revised.

Round 2

Reviewer 1 Report

Dear Authors,

You have adressed all my comments and corrections. The Manuscript can be published after minor Revision. 

Kind Regards